# Menstrual and Reproductive Factors for Gastric Cancer in Postmenopausal Women: The 2007–2020 Korea National Health and Nutrition Examination Survey

**DOI:** 10.3390/ijerph192114468

**Published:** 2022-11-04

**Authors:** Heekyoung Song, Jung Yoon Park, Ju Myung Song, Youngjae Yoon, Yong-Wook Kim

**Affiliations:** 1Department of Obstetrics and Gynecology, Incheon St. Mary’s Hospital, College of Medicine, The Catholic University of Korea, Seoul, Korea, 56, Dongsu-ro, Bupyeong-gu, Incheon 21431, Korea; 2Department of Obstetrics and Gynecology, Seoul St. Mary’s Hospital, College of Medicine, The Catholic University of Korea, Seoul, Korea, 222, Banpo-daero, Seocho-gu, Seoul 06591, Korea; 3Department of Surgery, Incheon St. Mary’s Hospital, College of Medicine, The Catholic University of Korea, Seoul, Korea, 56, Dongsu-ro, Bupyeong-gu, Incheon 21431, Korea; 4Department of Obstetrics and Gynecology, Eunpyeong St. Mary’s Hospital, College of Medicine, The Catholic University of Korea, Seoul, Korea, 1021, Tongil-ro, Eunpyeong-gu, Seoul 03312, Korea

**Keywords:** Korea National Health and Nutrition Examination Survey (KNHANES), stomach neoplasm, menarche, menopause, alcohol, myocardial infarction

## Abstract

Globally, the incidence of gastric cancer is lower in women than in men. It is thought that menstrual and reproductive factors may be related to their lower incidence of gastric cancer. This cross-sectional study examined menstrual, reproductive, and other factors in 20,784 postmenopausal women from the 2007–2020 Korea National Health and Nutrition Examination Survey (KNHANES). A univariate logistic regression analysis was performed, and then a multivariate logistic regression analysis for significant factors in the univariate analysis was conducted. In the multivariate logistic regression analysis, the age at menarche (odds ratio [OR] 1.08, 95% confidence interval [CI] 1.00−1.06, *p* = 0.035) and myocardial infarction (OR 2.43, 95% CI 1.05−5.62, *p* = 0.026) showed a significant association with increased incidence of gastric cancer. The age at menopause (OR 0.97, 95% CI 0.95−1.00, *p* = 0.03), the age at the first childbirth (OR 0.93, CI 0.89−0.97, *p* = 0.007), and the experience of alcohol consumption (OR 0.68, 95% CI 0.5–0.91, *p* = 0.003) showed a significant association with a decreased incidence of gastric cancer. Late menarche, early menopause, early aged first childbirth, and myocardial infarction are estimated to be risk factors for gastric cancer in postmenopausal Korean women.

## 1. Introduction

Gastric cancer is the 6th most common in both sexes, with 1,089,103 new diagnoses worldwide in 2020 [1,2]. The age-standardized incidence rate per 100,000 persons for gastric cancer was 30.4 in South Korea in 2018 [3]. The Korean National Health Insurance Service (NHIS) provides a national cancer screening program for the early detection of five major cancers (stomach, colon, liver, breast, and uterine cervix) in individuals over 40 years of age. This cancer screening program likely improves gastric cancer survival, as its 5-year relative survival rate improved from 43.8% in 1993–1995 to 77.0% in 2014–2018 [3]. Nevertheless, gastric cancer remained the most common cancer in both sexes in 2018 [3].

Gastric cancer is diagnosed at different incidences according to sex. In 2019, global incidence, death, and disability-adjusted life-years were higher among males than females [4]. In addition, males had higher aged-standardized mortality rates per 100,000 than females in Korea in 2018 (11 vs. 4.1) [3]. The difference in incidence between males and females was supposed to be because of the estrogen-protective effects. A longer period of fertility in females (the time from menarche to menopause) increases their lifetime exposure to endogenous estrogens. It has been reported in several studies that estrogen may reduce the risk of gastric cancer [5,6,7]. In molecular studies, estrogen decreased the progression of gastric cancer by inhibiting erbB-2 oncogene expression [8]. Additionally, 17ß-estradiol decreased migration in all epithelial cell lines, and this decrease was more profound in malignant cell lines [9]. Based on these studies, estrogen is thought to play an important role in preventing gastric cancer.

To confirm the relationship between menstrual and reproductive factors and gastric cancer, several retrospective cross-sectional and prospective cohort studies using national data have been performed in China, Canada, Japan, etc. [6,7,10]. We retrospectively reviewed data from the Korea National Health and Nutrition Examination Survey (KNHANES) to examine the relationship of menstrual, reproductive, and other factors with gastric cancer.

## 2. Materials and Methods

This cross-sectional study examined menstrual, reproductive, and other factors in 20,784 postmenopausal women using the KNHANES data collected between 2007 and 2020. To confirm the precise effect of menstrual and reproductive factors, only postmenopausal women who had finished all their reproductive events were included in this study. The Korea National Health and Nutrition Examination Survey began in 1998 and is administered by the Division of Health and Nutrition Survey under the Korea Disease Control and Prevention Agency [11]. It monitors the trends in health risk factors, assesses the prevalence of chronic diseases, and provides data for the development and evaluation of new and upcoming South Korean health policies and programs for healthcare professionals. Participants were selected using proportional allocation systematic sampling with multistage stratification. The health interviews and examinations were conducted in mobile examination centers, while the nutritional surveys were performed by trained physicians or nurses who visited each household. Written informed consent was acquired from all participants before the survey was administered. All data is available in the KNHANES database (https://knhanes.kdca.go.kr/knhanes/sub03/sub03_05.do (accessed on 20 May 2022)). Of the total 113,091 individuals surveyed using the KNHANES between 2007 and 2020, 51,551 males were excluded, and 36,774 premenopausal women and women under the age of 19 were excluded. 3982 postmenopausal women were excluded because of missing data. A total of 20,784 postmenopausal women were included in the final analysis (Figure 1). Of the total postmenopausal women, 185 were diagnosed with gastric cancer. The annual participant distribution is listed in Table 1.

Menstrual and reproductive factors like oral contraceptive use (for at least 1 month), age at menarche or menopause, age at the first or last childbirth, the number of pregnancies, and the breastfeeding experience (for at least 1 month) were analyzed in two groups of study participants: women without gastric cancer and women with gastric cancer. Additional analysis was performed to evaluate the association of alcohol consumption, smoking, and chronic diseases such as hypertension, diabetes mellitus, dyslipidemia, cerebrovascular accident, myocardial infarction, and angina pectoris. All participants were asked to submit responses to a categorized survey on alcohol consumption: experience (yes or no), frequency (once a month, two to four times per month, two or three times per week, more than four times a week), and amount (unit: glass). The survey on smoking was categorized as experience, frequency (smoking days per month), and amount (unit: a cigarette).

This study was approved by the Institutional Review Board of Incheon St. Mary’s Hospital, College of Medicine, Catholic University of Korea, Seoul, Korea (OC22ZISI0085). Statistical analyses were performed using the Stata (version 17; Stata Corp LLC, College Station, TX, USA), reflecting the complex sampling design of the KNHANES. 

The menstrual, reproductive, and other clinical factors of the participants, with and without gastric cancer, were compared using the independent *t*-tests for parametric continuous variables and Fisher’s exact probability tests for categorical variables. Statistical significance was set at *p* < 0.05. Logistic regression models were used to compare the participants with gastric cancer and those without gastric cancer to evaluate the estimated risk factors of gastric cancer. A univariate analysis was performed for factors with statistical significance in the independent t-test or Fisher’s exact test. Factors with statistical significance in the univariate analysis were included in the multivariate analysis. Odds ratios (OR) and 95% confidence intervals (CI) were calculated using the logistic regression models.

## 3. Results

### 3.1. General Characteristics of Participants

Considering general characteristics, the women diagnosed with gastric cancer were older in age (68.58 vs. 64.14 years) compared with those without gastric cancer. Women in the gastric cancer group had menarche at a later age (15.89 vs. 14.48 years), and reached menopause earlier (48.09 vs. 49.00 years). Additionally, the age at the first childbirth was earlier (22.96 vs. 23.85 years) in the gastric cancer group. The use of oral contraceptives, age at the last childbirth, frequency of pregnancy, and breast feeding experience were not significantly different between the two groups. Among the variables related to alcohol consumption and smoking history, lifetime smoking experience and smoking duration did not show significant differences between the two groups, while alcohol consumption experience (60.00% vs. 70.01%) was significantly lower in the gastric cancer group. Although drinking frequency, binge drinking frequency, and the amount of drinking are not presented in the table, there were no significant differences between the two groups. Regarding chronic disease history, myocardial infarction (3.24% vs. 1.14%) and angina pectoris (6.49% vs. 3.31%) was significantly higher in the gastric cancer group. However, dyslipidemia (14.59% vs. 27.46%) was significantly lower in the gastric cancer group (Table 2).

### 3.2. Menstrual and Reproductive Factors for Gastric Cancer

A univariate logistic analysis was conducted, followed by a multivariate analysis to evaluate the menstrual and reproductive factors for gastric cancer. The odds ratios of menstrual and reproductive factors are shown in Table 3. The OR of menarche age was 1.10 (95% CI 1.03−1.18, *p* = 0.006) in the univariate analysis and 1.08 (95% CI 1.00−1.06, *p* = 0.035) in the multivariate analysis. The OR of age at menopause was significant in all analyses (OR 0.97, 95% CI 0.94−0.99, *p* = 0.016 in the univariate analysis, OR 0.97, 95% CI 0.95−1.00, *p* = 0.03 in the multivariate analysis). As for the age at the first childbirth, the OR in the univariate analysis was 0.93 (95% CI 0.89−0.97, *p* = 0.001), and in the multivariate analysis it was 0.94 (95% CI 0.90−0.98, *p* = 0.007).

### 3.3. Alcohol Consumption, Smoking, and Chronic Diseases for Gastric Cancer

A univariate logistic regression analysis was conducted followed by a multivariate analysis for lifestyles and chronic diseases. The results of these analyses are shown in Table 3. Among alcohol consumption and smoking factors, only the experience of alcohol consumption was significant in both analyses (univariate: OR 0.64, 95% CI 0.48−0.86, *p* = 0.003; multivariate: OR 0.68, 95% CI 0.50−0.91, *p* = 0.003). Dyslipidemia was significantly associated with a decreased incidence of gastric cancer in the univariate analysis (OR 0.45, 95% CI 0.30−0.68, *p* < 0.001). However, it was not statistically significant in the multivariate analysis (OR 0.90, 95% CI 0.59−1.36, *p* = 0.473). Myocardial infarction was significantly associated with an increased incidence of gastric cancer in both analyses (univariate: OR 2.90, 95% CI 1.27−6.62, *p* = 0.011; multivariate: OR 2.43, 95% CI 1.05−5.62, *p* = 0.026), while angina pectoris was only significant in the univariate analysis (OR 2.03, 95% CI 1.12−3.66, *p* < 0.019).

## 4. Discussion

This large cross-sectional study sought to confirm the relationship between menstrual, reproductive, and other factors, and the risk of gastric cancer using Korean national data. Various studies have examined the relationship between sex hormones and gastric cancer, supported by the presence of steroid receptors in gastric cancer tissues and normal gastric mucosa [12,13]. It is estimated that gastric cancer is less common in women than in men because of the protective effect of estrogen against gastric cancer [14].

Considering the menstrual factors, according to the multivariate analysis conducted in this study, late menarche age and early menopause age showed a significant association with increased incidence of gastric cancer. Similar results were observed in a large-scale study using Canadian national data [7]. Rahman et al. reported that dysregulation of estrogen receptors is associated with gastric cancer [2]. In addition, a recent meta-analysis showed that the use of hormone replacement lowered the risk of gastric cancer, suggesting the cancer-growth inhibitory effect of estrogens [15]. Additionally, women who underwent ovariectomy had a 79% increased risk of gastric cancer (based on 25 cases) compared with women who did not (hazard ratio 1.79, 95% CI 1.15−2.78) [16]. This result indicates that increasing lifetime exposure to endogenous estrogens protects against gastric cancer [5,6,7].

In our multivariate analysis regarding reproductive factors, the early age of a woman’s first childbirth had a significant association with an increased incidence of gastric cancer. However, the use of oral contraceptives, the age at the last childbirth, the frequency of pregnancy, and the experience of breast feeding, showed no statistical significance. In the study using Canadian national data, compared with nulliparity, four or more births were associated with a decreased risk for gastric cancer, as was being pregnant for five months or longer if the first pregnancy occurred at younger than 24 years (OR 0.55, 95% CI 0.31−0.96) or at 25 years of age or older (OR 0.67, 95% CI 0.38−1.18) [7]. In a Japanese national prospective study for postmenopausal women, the age at the first childbirth showed no statistical significance [10]. These discordances demonstrate a complicated relationship between reproductive factors and the occurrence of gastric cancer. Furthermore, some studies suggested that the association of gastric cancer with different patterns of *Helicobacter pylori* infection during lifetime [17,18,19]. Larger prospective studies are needed to clarify the association of these factors with gastric cancer.

Among alcohol consumption and smoking history, the experience of alcohol consumption was related to a decreased occurrence of gastric cancer in our multivariate analysis. The experience and period of smoking showed no statistical significance in this study. Several studies have shown that heavy alcohol consumption and smoking are risk factors for gastric cancer [20,21,22]. However, in a Korean prospective cohort study, no significant association was found between smoking habits and gastric cancer survival, while light alcohol consumption was found to be associated with a significantly lower risk of gastric cancer death (HR 0.52, 95% CI 0.36−0.75 for <20 g/day for women or <40 g/day for men vs. never and past consumption) [23]. The reason for such different results is presumed to be that the effects of drinking and smoking on gastric cancer in women and men are different because of differences in body characteristics and lifestyles. These results may also differ according to race or country.

Hypertension, diabetes mellitus, and stroke, showed no statistical significance in Fisher’s exact test. Dyslipidemia was significantly associated with the decreased incidence of gastric cancer in the univariate analysis. However, it was not statistically significant in the multivariate analysis. Only myocardial infarction was significantly associated with an increased incidence of gastric cancer in both analyses. In various studies, the incidence rate of arterial thromboembolism in patients with cancer was demonstrated to be higher than in patients without cancer, and this event was related to postoperative complications [24,25]. Furthermore, the most common event of cancer-related thromboembolism has been reported to be deep vein thrombosis, and not arterial thrombosis [26,27]. Additionally, stroke was reported as the most common complication of arterial thrombosis [26,27]. In this study, stroke was not a significant factor for gastric cancer (Table 1), even though myocardial infarction was a significant factor for gastric cancer (Table 3). The relationship between myocardial infarction and gastric cancer may result from mechanisms other than thromboembolism. Further studies are needed to confirm these results.

This study had several limitations. The study was cross-sectional; thus, causal relationships could not be analyzed. Additionally, the data were derived from self-reported questionnaires based on participant recall, and some parts of the questionnaires were missing answers. Nevertheless, this study was conducted on a large scale (*n* = 20,784), using a national dataset over a relatively long period, which is considered a major strength of this study.

## 5. Conclusions

Late menarche, early menopause, early aged first childbirth, and myocardial infarction, are estimated to be risk factors for gastric cancer in postmenopausal Korean women.

## Figures and Tables

**Figure 1 ijerph-19-14468-f001:**
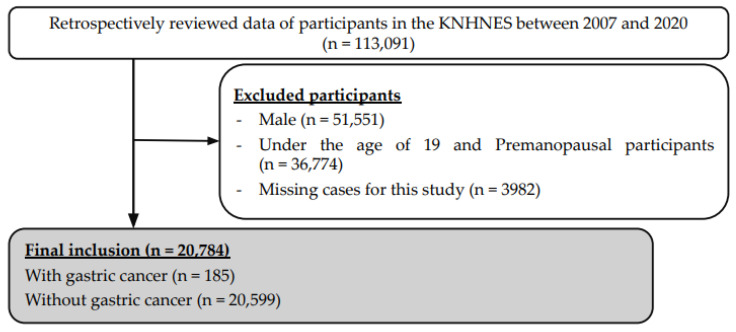
Flow chart for selection of the study population. Abbreviation: KNHANES, Korea National Health and Nutrition Examination Survey.

**Table 1 ijerph-19-14468-t001:** Sample distribution by study year.

Year	Without Gastric Cancer (*n*)	With Gastric Cancer (*n*)	Gastric Cancer (%)
2007	680	4	0.58
2008	1305	7	0.53
2009	1641	14	0.85
2010	1632	13	0.79
2011	1705	11	0.64
2012	1622	15	0.92
2013	1419	17	1.18
2014	1416	20	1.39
2015	1439	16	1.10
2016	1587	11	0.69
2017	1649	18	1.08
2018	1603	12	0.74
2019	1547	16	1.02
2020	1354	11	0.81
Total	20,599	185	0.89

**Table 2 ijerph-19-14468-t002:** General characteristics of participants.

Variables	With Gastric Cancer	Without Gastric Cancer	Statistic	
	*n* (%) or (M ± SD)	*n* (%) or (M ± SD)	χ^2^ or *t*-Test	*p*-Value
Age (M ± SD)	68.58 ± 9.53	64.14 ± 9.15	−6.57	<0.001
Menstrual and reproductive history				
Use of the oral contraceptives (*n*, %)	45 (24.32)	4543 (22.05)	0.55	0.459
Age at menarche (M ± SD)	15.89 ± 2.11	14.48 ± 2.02	−2.74	0.006
Age at menopause (M ± SD)	48.09 ± 5.59	49.00 ± 5.09	2.41	0.016
Age at the first childbirth (M ± SD)	22.96 ± 3.33	23.85 ± 3.59	3.31	<0.001
Age at the last childbirth (M ± SD)	29.83 ± 4.31	29.70 ± 4.49	−0.36	0.718
Frequency of pregnancy (M ± SD)	4.90 ± 2.35	4.66 ± 2.27	−1.43	0.154
Breast feeding experience (*n*, %)	149 (94.90)	15,176 (90.27)	3.82	0.051
Alcohol consumption and smoking history				
Experience of alcohol consumption (*n*, %)	111 (60.00)	14,422 (70.01)	8.74	0.003
Experience of smoking (*n*, %)	18 (9.73)	1548 (7.52)	1.28	0.257
Period of smoking (M ± SD)	23.44 ± 89.86	16.90 ± 78.63	−1.13	0.261
Chronic disease history				
Hypertension (*n*, %)	78 (42.16)	8345 (40.51)	0.21	0.649
Diabetes mellitus (*n*, %)	28 (15.14)	2938 (14.26)	0.11	0.736
Dyslipidemia (n, %)	27 (14.59)	5657 (27.46)	15.28	<0.001
Stroke (*n*, %)	7 (3.78)	694 (3.37)	0.10	0.756
Myocardial infarction (*n*, %)	6 (3.24)	235 (1.14)	7.07	0.008
Angina pectoris (*n*, %)	12 (6.49)	681 (3.31)	5.75	0.016

Abbreviation: M, mean; SD, standard deviation. The clinical features were compared using either Fisher’s exact probability tests (χ^2^) or independent *t*-tests. Statistical significance was set at *p* < 0.05.

**Table 3 ijerph-19-14468-t003:** Logistic regression analyses for menstrual, reproductive, and other factors.

	Univariate		Multivariate	
	OR (95% CI)	*p*-Value	OR (95% CI)	*p*-Value
Menstrual and reproductive history				
**Age at menarche**	1.10 (1.03−1.18)	0.006	1.08 (1.00−1.06)	0.035
**Age at menopause**	0.97 (0.94−0.99)	0.016	0.97 (0.95−1.00)	0.030
**Age at the first childbirth**	0.93 (0.89−0.97)	0.001	0.94 (0.90−0.98)	0.007
Alcohol consumption and smoking history				
**Experience of alcohol consumption**	0.64 (0.48−0.86)	0.003	0.68 (0.50−0.91)	0.003
Chronic disease history				
**Dyslipidemia**	0.45 (0.30−0.68)	<0.001	0.90 (0.59−1.36)	0.473
**Myocardial Infarction**	2.90 (1.27−6.62)	0.011	2.43 (1.05−5.62)	0.026
**Angina pectoris**	2.03 (1.12−3.66)	0.019	1.76 (0.96−3.21)	0.058

Total number = 20,784. Abbreviation: OR, odd ratio; CI, confidence interval. Statistical significance was set at *p* < 0.05.

## Data Availability

Data were obtained from KCDC and are available from https://knhanes.kdca.go.kr/knhanes/sub03/sub03_05.do (accessed on 20 May 2022).

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
