# Peer review of "Menstrual and Reproductive Factors for Gastric Cancer in Postmenopausal Women: The 2007–2020 Korea National Health and Nutrition Examination Survey"

_ijerph, 2022, doi:10.3390/ijerph192114468_

Round 1
Reviewer 1 Report
The article entitled “The Association of Menstrual and Reproductive Factors with Gastric Cancer in Women: The 2007-2020 Korea National Health and Nutrition Examination Survey” represents an interesting original scientific paper examining the relationship of menstrual, reproductive, and other factors with gastric cancer in the Korean female population.
The entire manuscript is well written. However, some obstacles must be removed before the article is ready for publication.
These, among others, include:
The model 1 and 2 of multivariate (adjusted) logistic regression analysis should be more concisely explained in the Materials and method section.
The authors should explain whether there was any correction (e.g., Bonferroni procedure or like) for multiple comparisons applied.
A minor revision of the manuscript is recommended.
Author Response
Thank you so much for your precise comments on the manuscript. We believe the issues pointed out by you are a very important part that we missed in this study. In this revision, we have improved the paper by addressing the issues raised by the review team. We sincerely thank you for giving us another opportunity to revise the paper. The revised sentences are highlighted in blue, and the added sentences are highlighted in yellow. Also, to increase the accuracy of the statistics, we only performed statistical analysis on postmenopausal women who had completed all reproductive activities as suggested by Reviewer 2. We have proofread the entire manuscript again to improve its readability. We hope that we have addressed the review team’s concerns adequately.
1) The model 1 and 2 of multivariate (adjusted) logistic regression analysis should be more concisely explained in the Materials and method section.
Response: Thank you for your valuable suggestion. We reanalyzed the data and simplified the statistical methods. We compared the data using the independent t-test or Fisher’s exact test and performed logistic regression using univariate and multivariate analyses. We have revised the related sentences in the Materials and Method as follows: “The menstrual, reproductive, and other clinical factors of the participants with and without gastric cancer were compared using the independent t-tests for parametric continuous variables and Fisher’s exact probability tests for categorical variables. Statistical significance was set at p<0.05. Logistic regression models were used to compare the participants with gastric cancer and those without gastric cancer to evaluate the estimated risk factors of gastric cancer. A univariate analysis was performed for factors with statistical significance in the independent t-test or Fisher’s exact test. Factors with statistical significance in the univariate analysis were included in the multivariate analysis. Odds ratios (OR) and 95% confidence intervals (CI) were calculated using the logistic regression models.
”
2) The authors should explain whether there was any correction (e.g., Bonferroni procedure or like) for multiple comparisons applied.
Response: Thank you for the valuable suggestion. We analyzed all the factors between the gastric cancer and non-gastric cancer groups using the Fisher’s exact probability test or independent t-tests. Bonferroni procedure or ANOVA is used to compare variables among more than three groups. Therefore, we revised the sentences as follows: “The menstrual, reproductive, and other clinical factors of the participants with and without gastric cancer were compared using the independent t-tests for parametric continuous variables and Fisher’s exact probability tests for categorical variables. Statistical significance was set at p<0.05. “

Reviewer 2 Report
The paper submitted by Song et al. aims to verify if difference in women reproductive histories, behavioral factors and chronic diseases could be associated to gastric cancer in South Korean women.
The authors based their statistical analysis on data extracted from the Korea National Health and Nutrition Examination Survey (KNHANES) between 2007 and 2020 for a total of 49671 women of which 236 affected by gastric cancer.
The authors implemented standard statistical techniques such as descriptive statistics, univariate and multivariate analysis to study these variables in relation to gastric cancer.
The conclusions reached by the authors are that late menarche, early menopause, myocardial infarction, and angina pectoris could be risk factors for gastric cancer in Korean women.
The idea underlying the research is interesting but the paper presents some weaknesses that need to be explained and overcome.
In my opinion, the main weakness of the paper concerns the choice of the variables introduced in the analysis. In fact the authors have considered the whole sample of women regardless of their age (19 years or older) but some variables are highly dependent on woman age, such as “age at last childbirth”, “experience of pregnancy” and “frequency of pregnancy”.
These variables cannot be calculated without introducing a bias in the analysis if women who are still in their reproductive period and women who are already in the post-reproductive period are both included in sample. To avoid the bias, the authors have different solutions such as:
1) grouping women by age classes and analyze all the variables inside each class, or
2) selecting only women in post-reproductive period and using all the variables, or
3) considering the whole sample but choosing only the variables that do not depend on age.
In tables 2, 3, 4 only the total number of women included in the analysis was reported but no data regarding how many women with or without cancer are presented. As the total number of women varied among variables, due to missing data, and gastric cancer has a low incidence in the sample, it would be important to have information on sample sizes on which the statistical analysis was performed.
The statistical analysis carried out on the data was not well explained in the “Materials and Methods” section (84-97 rows). The difference between model 1 and model 2 and the variables involved in multivariate analysis is not clear and this does not help to understand correctly the results.
Furthermore, the results presented in tables 2, 3 and 4 have no information on which of the two models explains the greater variability of the data. The two models give different results and there is no statistical information on how to interpret them and which of the two is the most performing.
In view of the above considerations, I think the current manuscript cannot be published as it is. I suggested to the authors major revisions and new sampling strategies to analyze the correct variables weighted by woman age. If this further analysis are positive the paper can be reconsidered and may be suitable to publication.
Author Response
Thank you for your precise comments on the manuscript. As per your suggestions, we made the following changes:
- We reanalyzed the data of postmenopausal women only.
- Also, we excluded the participants with missing data (n = 3,982).
- Finally, we inserted statistical information below tables 2 and 3.
We revised the sentences in materials and methods and inserted “Of the total 113,091 individuals surveyed using the KNHANES between 2007 and 2020, 51,551 males were excluded, and 36,774 premenopausal women and women under the age of 19 were excluded. 3,982 postmenopausal women were excluded because of missing data. A total of 20,784 postmenopausal women were included in the final analysis (Figure 1). Of the total postmenopausal women, 185 were diagnosed with gastric cancer. The annual participant distribution is listed in Table 1.”
Figure 1. Flowchart for selection of the study population.
Abbreviation: KNHANES, Korea National Health and Nutrition Examination Survey
Furthermore, we revised all the results using new statistical analyses as follows:
3.1. General Characteristics of Participants
Considering general characteristics, the women diagnosed with gastric cancer were older in age (68.58 vs. 64.14 years) compared with those without gastric cancer. Women in the gastric cancer group had menarche at a later age (15.89 vs. 14.48 years), and reached menopause earlier (48.09 vs. 49.00 years). Additionally, the age at the first childbirth was earlier (22.96 vs. 23.85 years) in the gastric cancer group. The use of the oral contraceptives, age at the last childbirth, frequency of pregnancy, and breast feeding experience were not significantly different between the two groups. Among the variables related to alcohol consumption and smoking history, lifetime smoking experience and smoking duration did not show significant differences between the two groups, while alcohol consumption experience (60.00% vs. 70.01%) was significantly lower in the gastric cancer group. Although drinking frequency, binge drinking frequency, and the amount of drinking are not presented in the table, there were no significant differences between the two groups. Regarding chronic disease history, myocardial infarction (3.24% vs. 1.14%) and angina pectoris (6.49% vs. 3.31%) was significantly higher in the gastric cancer group. However, dyslipidemia (14.59% vs. 27.46%) was significantly lower in the gastric cancer group (Table 2).
Table 2. General characteristics of participants.
|
Variables
|
With |
Without gastric cancer |
Statistic |
||||
|
n (%) or (M±SD) |
n (%) or (M±SD) |
χ2 or t-test |
p-value |
||||
|
Age (M ± SD) |
68.58 ± 9.53 |
64.14 ± 9.15 |
-6.57 |
<0.001 |
|||
|
Menstrual and reproductive history |
|||||||
|
Use of the oral contraceptives (n, %) |
45 (24.32) |
4,543 (22.05) |
0.55 |
0.459 |
|||
|
Age at menarche (M ± SD) |
15.89 ± 2.11 |
14.48 ± 2.02 |
-2.74 |
0.006 |
|||
|
Age at menopause (M ± SD) |
48.09 ± 5.59 |
49.00 ± 5.09 |
2.41 |
0.016 |
|||
|
Age at the first childbirth (M ± SD) |
22.96 ± 3.33 |
23.85 ± 3.59 |
3.31 |
<0.001 |
|||
|
Age at the last childbirth (M ± SD) |
29.83 ± 4.31 |
29.70 ± 4.49 |
-0.36 |
0.718 |
|||
|
Frequency of pregnancy (M ± SD) |
4.90 ± 2.35 |
4.66 ± 2.27 |
-1.43 |
0.154 |
|||
|
Breast feeding experience (n, %) |
149 (94.90) |
15,176 (90.27) |
3.82 |
0.051 |
|||
|
Alcohol consumption and smoking history |
|||||||
|
Experience of alcohol consumption (n, %) |
111 (60.00) |
14,422 (70.01) |
8.74 |
0.003 |
|||
|
Experience of smoking (n, %) |
18 (9.73) |
1,548 (7.52) |
1.28 |
0.257 |
|||
|
Period of smoking (M ± SD) |
23.44 ± 89.86 |
16.90 ± 78.63 |
-1.13 |
0.261 |
|||
|
Chronic disease history |
|||||||
|
Hypertension (n, %) |
78 (42.16) |
8,345 (40.51) |
0.21 |
0.649 |
|||
|
Diabetes mellitus (n, %) |
28 (15.14) |
2,938 (14.26) |
0.11 |
0.736 |
|||
|
Dyslipidemia (n, %) |
27 (14.59) |
5,657 (27.46) |
15.28 |
<0.001 |
|||
|
Stroke (n, %) |
7 (3.78) |
694 (3.37) |
0.10 |
0.756 |
|||
|
Myocardial infarction (n, %) |
6 (3.24) |
235 (1.14) |
7.07 |
0.008 |
|||
|
Angina pectoris (n, %) |
12 (6.49) |
681 (3.31) |
5.75 |
0.016 |
|||
Abbreviation: M, mean; SD, standard deviation. The clinical features were compared using either Fisher’s exact probability tests (χ2) or independent t-tests. Statistical significance was set at p<0.05.
3.2. Menstrual and reproductive factors for gastric cancer
A univariate logistic analysis was conducted, and followed by a multivariate analysis to evaluate the menstrual and reproductive factors for gastric cancer. Odds ratios of menstrual and reproductive factors are shown in Table 3. The OR of menarche age was 1.10 (95% CI 1.03−1.18, p=0.006) in the univariate analysis and 1.08 (95% CI 1.00−1.06, p=0.035) in the multivariate analysis. The OR of age at menopause was significant in all analyses (OR 0.97, 95% CI 0.94−0.99, p=0.016 in the univariate analysis, OR 0.97, 95% CI 0.95−1.00, p=0.03 in the multivariate analysis). As for the age at the first childbirth, the OR in the univariate analysis was 0.93 (95% CI 0.89−0.97, p=0.001), and that in the multivariate analysis was 0.94 (95% CI 0.90−0.98, p=0.007).
3.3. Alcohol consumption, smoking, and chronic diseases for gastric cancer
A univariate logistic regression analysis was conducted followed by a multivariate analysis for lifestyles and chronic diseases. The results of these analyses are shown in Table 3. Among alcohol consumption and smoking factors, only the experience of alcohol consumption was significant in the both analyses (univariate: OR 0.64, 95% CI 0.48−0.86, p=0.003; multivariate: OR 0.68, 95% CI 0.50−0.91, p=0.003). Dyslipidemia was significantly associated with decreased incidence of gastric cancer in the univariate analysis (OR 0.45, 95% CI 0.30−0.68, p<0.001). However, it was not statistically significant in the multivariate analysis (OR 0.90, 95% CI 0.59−1.36, p=0.473). Myocardial infarction was significantly associated with increased incidence of gastric cancer in the both analyses (univariate: OR 2.90, 95% CI 1.27−6.62, p=0.011; multivariate: OR 2.43, 95% CI 1.05−5.62, p=0.026), while angina pectoris was only significant in the univariate analysis (OR 2.03, 95% CI 1.12−3.66, p<0.019).
Table 3. Logistic regression analyses for menstrual, reproductive, and other factors.
|
|
Univariate |
Multivariate |
|||
|
OR (95% CI) |
p-value |
OR (95% CI) |
p-value |
||
|
Menstrual and reproductive history |
|||||
|
Age at menarche |
1.10 (1.03−1.18) |
0.006 |
1.08 (1.00−1.06) |
0.035 |
|
|
Age at menopause |
0.97 (0.94−0.99) |
0.016 |
0.97 (0.95−1.00) |
0.030 |
|
|
Age at the first childbirth |
0.93 (0.89−0.97) |
0.001 |
0.94 (0.90−0.98) |
0.007 |
|
|
Alcohol consumption and smoking history |
|||||
|
Experience of alcohol consumption |
0.64 (0.48−0.86) |
0.003 |
0.68 (0.50−0.91) |
0.003 |
|
|
Chronic disease history |
|||||
|
Dyslipidemia |
0.45 (0.30−0.68) |
<0.001 |
0.90 (0.59−1.36) |
0.473 |
|
|
Myocardial Infarction |
2.90 (1.27−6.62) |
0.011 |
2.43 (1.05−5.62) |
0.026 |
|
|
Angina pectoris |
2.03 (1.12−3.66) |
0.019 |
1.76 (0.96−3.21) |
0.058 |
|
.
Total number = 20,784, Abbreviation: OR, odd ratio; CI, confidence interval. Statistical significance was set at p<0.05.
We have proofread the entire manuscript again to improve readability. We hope that we have addressed the review team’s concerns adequately.

Reviewer 3 Report
Dear authors, the manuscript focuses on the important Association of Menstrual and Reproductive Factors with Gastric Cancer in Women
Please see some comments to improve the quality of the paper.
Introduction: This section is well written. May be you can quote the countries where similar studies have been conducted, so easy for the reader to relate.
Methods:
Please elaborate on the study design,
how the data was collected?
please describe the study tool/data collection form
please describe the the statistical tests used.
Results: Please state the type of statistical test used under the tables.
Author Response
Thank you for your precise comments on our manuscript. In this revision, we have improved the paper by addressing the remaining issues raised by the review team. We sincerely thank you for giving us another opportunity to revise the paper. The revised sentences are highlighted in blue, and the added sentences are highlighted in yellow. Also, to increase the accuracy of the statistics, we only performed statistical analysis on postmenopausal women who had completed all reproductive activities as suggested by Reviewer 2. Furthermore, we proofread the entire manuscript to improve its readability. We hope that we have addressed the review team’s concerns adequately.
1) Introduction: This section is well written. Maybe you can quote the countries where similar studies have been conducted, so easy for the reader to relate.
Response: Thank you for the valuable suggestion. We inserted a new sentence and revised several sentences in the Introduction as follows: “To confirm the relationship between menstrual and reproductive factors and gastric cancer, several retrospective cross-sectional and prospective cohort studies using national data have been performed in China, Canada, Japan, etc [6,7,10]. We retrospectively reviewed data from the Korea National Health and Nutrition Examination Survey (KNHANES) to examine the relationship of menstrual, reproductive, and other factors with gastric cancer.
“
2) Methods:
Please elaborate on the study design, how the data was collected?
Response: This cross-sectional study examined menstrual, reproductive, and other factors in 20,784 postmenopausal women using the KNHANES data collected between 2007 and 2020. To confirm precise effect of menstrual and reproductive factors, only postmenopausal women who had finished all their reproductive events were included in this study. The Korea National Health and Nutrition Examination Survey began in 1998 and is administered by the Division of Health and Nutrition Survey under the Korea Disease Control and Prevention Agency [11]. It monitors the trends in health risk factors, assesses the prevalence of chronic diseases, and provides data for the development and evaluation of new and upcoming South Korean health policies and programs for healthcare professionals. Participants were selected using proportional allocation systematic sampling with multistage stratification. The health interviews and examinations were conducted in mobile examination centers, while the nutritional surveys were performed by trained physicians or nurses who visited each household. Written informed consent was acquired from all participants before the survey was administered. All data is available in the KNHANES database (https://knhanes.kdca.go.kr/knhanes/sub03/sub03_05.do). Of the total 113,091 individuals surveyed using the KNHANES between 2007 and 2020, 51,551 males were excluded, and 36,774 premenopausal women and women under the age of 19 were excluded. 3,982 postmenopausal women were excluded because of missing data. A total of 20,784 postmenopausal women were included in the final analysis (Figure 1). Of the total postmenopausal women, 185 were diagnosed with gastric cancer. The annual participant distribution is listed in Table 1.
please describe the study tool/data collection form.
Response: We have revised the Materials and methods section as follows:
“Menstrual and reproductive factors like oral contraceptive use (for at least 1 month), age at menarche or menopause, age at the first or last childbirth, the number of pregnancies, and the breastfeeding experience (for at least 1 month) were analyzed in two groups of study participants: women without gastric cancer and women with gastric cancer. Additional analysis was performed to evaluate the association of alcohol consumption, smoking, and chronic diseases such as hypertension, diabetes mellitus, dyslipidemia, cerebrovascular accident, myocardial infarction, and angina pectoris. All participants were asked to submit responses to a categorized survey on alcohol consumption: experience (yes or no), frequency (once a month, two to four times per month, two or three times per week, more than four times a week), and amount (unit: glass). The survey on smoking was categorized as experience, frequency (smoking days per month), and amount (unit: a cigarette). “
”
please describe the statistical tests used.
Response: We revised the initial sentences and added a new sentence to explain our statistical methods in the Materials and Methods section.
“The menstrual, reproductive, and other clinical factors of the participants with and without gastric cancer were compared using the independent t-tests for parametric continuous variables and Fisher’s exact probability tests for categorical variables. Statistical significance was set at p<0.05. Logistic regression models were used to compare the participants with gastric cancer and those without gastric cancer to evaluate the estimated risk factors of gastric cancer. A univariate analysis was performed for factors with statistical significance in the independent t-test or Fisher’s exact test. Factors with statistical significance in the univariate analysis were included in the multivariate analysis. Odds ratios (OR) and 95% confidence intervals (CI) were calculated using the logistic regression models. “
3) Results: Please state the type of statistical test used under the tables.
Response: We inserted types of statistical test below tables 2 and 3.
Table 2. General characteristics of participants.
|
Variables
|
With |
Without gastric cancer |
Statistic |
||||
|
n (%) or (M±SD) |
n (%) or (M±SD) |
χ2 or t-test |
p-value |
||||
|
Age (M ± SD) |
68.58 ± 9.53 |
64.14 ± 9.15 |
-6.57 |
<0.001 |
|||
|
Menstrual and reproductive history |
|||||||
|
Use of the oral contraceptives (n, %) |
45 (24.32) |
4,543 (22.05) |
0.55 |
0.459 |
|||
|
Age at menarche (M ± SD) |
15.89 ± 2.11 |
14.48 ± 2.02 |
-2.74 |
0.006 |
|||
|
Age at menopause (M ± SD) |
48.09 ± 5.59 |
49.00 ± 5.09 |
2.41 |
0.016 |
|||
|
Age at the first childbirth (M ± SD) |
22.96 ± 3.33 |
23.85 ± 3.59 |
3.31 |
<0.001 |
|||
|
Age at the last childbirth (M ± SD) |
29.83 ± 4.31 |
29.70 ± 4.49 |
-0.36 |
0.718 |
|||
|
Frequency of pregnancy (M ± SD) |
4.90 ± 2.35 |
4.66 ± 2.27 |
-1.43 |
0.154 |
|||
|
Breast feeding experience (n, %) |
149 (94.90) |
15,176 (90.27) |
3.82 |
0.051 |
|||
|
Alcohol consumption and smoking history |
|||||||
|
Experience of alcohol consumption (n, %) |
111 (60.00) |
14,422 (70.01) |
8.74 |
0.003 |
|||
|
Experience of smoking (n, %) |
18 (9.73) |
1,548 (7.52) |
1.28 |
0.257 |
|||
|
Period of smoking (M ± SD) |
23.44 ± 89.86 |
16.90 ± 78.63 |
-1.13 |
0.261 |
|||
|
Chronic disease history |
|||||||
|
Hypertension (n, %) |
78 (42.16) |
8,345 (40.51) |
0.21 |
0.649 |
|||
|
Diabetes mellitus (n, %) |
28 (15.14) |
2,938 (14.26) |
0.11 |
0.736 |
|||
|
Dyslipidemia (n, %) |
27 (14.59) |
5,657 (27.46) |
15.28 |
<0.001 |
|||
|
Stroke (n, %) |
7 (3.78) |
694 (3.37) |
0.10 |
0.756 |
|||
|
Myocardial infarction (n, %) |
6 (3.24) |
235 (1.14) |
7.07 |
0.008 |
|||
|
Angina pectoris (n, %) |
12 (6.49) |
681 (3.31) |
5.75 |
0.016 |
|||
Abbreviation: M, mean; SD, standard deviation. The clinical features were compared using either Fisher’s exact probability tests (χ2) or independent t-tests. Statistical significance was set at p<0.05.
Table 3. Logistic regression analyses for menstrual, reproductive, and other factors.
|
|
Univariate |
Multivariate |
|||
|
OR (95% CI) |
p-value |
OR (95% CI) |
p-value |
||
|
Menstrual and reproductive history |
|||||
|
Age at menarche |
1.10 (1.03−1.18) |
0.006 |
1.08 (1.00−1.06) |
0.035 |
|
|
Age at menopause |
0.97 (0.94−0.99) |
0.016 |
0.97 (0.95−1.00) |
0.030 |
|
|
Age at the first childbirth |
0.93 (0.89−0.97) |
0.001 |
0.94 (0.90−0.98) |
0.007 |
|
|
Alcohol consumption and smoking history |
|||||
|
Experience of alcohol consumption |
0.64 (0.48−0.86) |
0.003 |
0.68 (0.50−0.91) |
0.003 |
|
|
Chronic disease history |
|||||
|
Dyslipidemia |
0.45 (0.30−0.68) |
<0.001 |
0.90 (0.59−1.36) |
0.473 |
|
|
Myocardial Infarction |
2.90 (1.27−6.62) |
0.011 |
2.43 (1.05−5.62) |
0.026 |
|
|
Angina pectoris |
2.03 (1.12−3.66) |
0.019 |
1.76 (0.96−3.21) |
0.058 |
|
Total number = 20,784, Abbreviation: OR, odd ratio; CI, confidence interval. Statistical significance was set at p<0.05.

Round 2
Reviewer 2 Report
I have read the revised paper by Song et al. The authors carefully considered my suggestions, especially the most important one concerning the sample of women to be analysed in order to have a homogeneous sample that they did not have in the first version. They have limited the sample to women in post reproductive period and performed a new statistical analysis on its sample.
Considering the changes made by the authors, I think this paper can now be published in this journal.